# Maternal High-Fat Diet Promotes Abdominal Aortic Aneurysm Expansion in Adult Offspring by Epigenetic Regulation of IRF8-Mediated Osteoclast-like Macrophage Differentiation

**DOI:** 10.3390/cells10092224

**Published:** 2021-08-27

**Authors:** Makoto Saburi, Hiroyuki Yamada, Naotoshi Wada, Shinichiro Motoyama, Takeshi Sugimoto, Hiroshi Kubota, Daisuke Miyawaki, Noriyuki Wakana, Daisuke Kami, Takehiro Ogata, Satoaki Matoba

**Affiliations:** 1Department of Cardiovascular Medicine, Graduate School of Medical Science, Kyoto Prefectural University of Medicine, Kyoto 602-8566, Japan; msaburi@koto.kpu-m.ac.jp (M.S.); wada-n@koto.kpu-m.ac.jp (N.W.); motoyama@koto.kpu-m.ac.jp (S.M.); sugimoto@koto.kpu-m.ac.jp (T.S.); kbt-h@koto.kpu-m.ac.jp (H.K.); torisan@koto.kpu-m.ac.jp (D.M.); nw0920@koto.kpu-m.ac.jp (N.W.); matoba@koto.kpu-m.ac.jp (S.M.); 2Department of Regenerative Medicine, Graduate School of Medical Science, Kyoto Prefectural University of Medicine, Kyoto 602-8566, Japan; dkami@koto.kpu-m.ac.jp; 3Department of Pathology and Cell Regulation, Graduate School of Medical Science, Kyoto Prefectural University of Medicine, Kyoto 602-8566, Japan; ogatat@koto.kpu-m.ac.jp

**Keywords:** maternal high-fat diet, abdominal aortic aneurysm, MMP, osteoclast-like macrophages, NFATc1, TRAP, IRF8, histone modification, H3K27me3, EZH2

## Abstract

Maternal high-fat diet (HFD) modulates vascular remodeling in adult offspring. Here, we investigated the impact of maternal HFD on abdominal aortic aneurysm (AAA) development. Female wild-type mice were fed an HFD or normal diet (ND). AAA was induced in eight-week-old pups using calcium chloride. Male offspring of HFD-fed dams (O-HFD) showed a significant enlargement in AAA compared with the offspring of ND-fed dams (O-ND). Positive-staining cells for tartrate-resistant acid phosphate (TRAP) and matrix metalloproteinase (MMP) activity were significantly increased in O-HFD. The pharmacological inhibition of osteoclastogenesis abolished the exaggerated AAA development in O-HFD. The in vitro tumor necrosis factor-α-induced osteoclast-like differentiation of bone marrow-derived macrophages showed a higher number of TRAP-positive cells and osteoclast-specific gene expressions in O-HFD. Consistent with an increased expression of nuclear factor of activated T cells 1 (NFATc1) in O-HFD, the nuclear protein expression of interferon regulatory factor 8 (IRF8), a transcriptional repressor, were much lower, with significantly increased H3K27me3 marks at the promoter region. The enhancer of zeste homolog 2 inhibitor treatment restored IRF8 expression, resulting in no difference in NFATc1 and TRAP expressions between the two groups. Our findings demonstrate that maternal HFD augments AAA expansion, accompanied by exaggerated osteoclast-like macrophage accumulation, suggesting the possibility of macrophage skewing via epigenetic reprogramming.

## 1. Introduction

Maternal overnutrition and obesity during pregnancy and lactation have been well-recognized to increase the risk of cardiometabolic disorders throughout the lifespan of offspring [1,2,3]. The nutritional profile during fetal development has been shown to modify the gene expression in offspring tissues without changes in the DNA sequence, which is commonly referred to as “epigenetic reprogramming” [4,5,6]. Epigenetic modifications have emerged as a novel therapeutic approach for cardiovascular diseases (CVDs), as well as metabolic disorders and malignancies [7,8,9]. In earlier studies, we have shown that maternal high-fat diet (HFD) intake leads to atherosclerosis development and HFD-induced insulin resistance in offspring via the augmented macrophage-mediated inflammatory response [10,11]; however, the underlying epigenetic mechanisms have not been fully elucidated.

Abdominal aortic aneurysm (AAA) is one of the most prevalent atherosclerotic diseases globally and has a relatively high mortality, especially in elderly populations who eventually experience aortic rupture [12]. Although extensive efforts have been made to develop a therapeutic approach for preventing the annual expansion of AAA, an effective therapeutic strategy has not yet been devised [13,14]. Given that AAA development is closely related with the augmented accumulation of macrophages and consequent inflammatory response in the adventitia [15,16], it is likely that the epigenetic reprogramming of monocytes/macrophages via maternal HFD plays a critical role in AAA development in the offspring.

In this study, we examined the impact of maternal HFD on AAA development in offspring and investigated epigenetic mechanisms to modulate macrophage activity via maternal HFD. Maternal HFD augmented AAA development in adult offspring, along with the enhanced accumulation of osteoclast-like macrophages in the adventitia. Treatment with zoledronic acid (ZA) eliminated the exaggerated AAA development caused by maternal HFD intake. In vitro osteoclast differentiation from bone marrow-derived macrophages (BMDMs) was significantly enhanced, accompanied by the increased expression of NFATc1, a master regulator of osteoclast differentiation. Furthermore, the expression of interferon regulatory factor 8 (IRF8), a negative regulator of NFATc1 activity, was decreased in BMDMs, along with an increase in H3K27me3 marks at the IRF8 promoter region. Our findings suggest that the maternal HFD-induced reprogramming of macrophages in the offspring contributes to the enhanced development of AAA and that therapeutic targeting of the epigenetic modifications in the macrophage phenotype could potentially remediate and prevent AAA development.

## 2. Materials and Methods

### 2.1. Experimental Animals

Wild-type mice (C57BL/6N) were obtained from Shimizu Laboratory Supplies Co., Ltd. (Kyoto, Japan). Eight-week-old female mice were maintained on a normal diet (12.0% fat, 28.9% protein, and 59.1% carbohydrate; Oriental Yeast Co., Tokyo, Japan) (ND) or high-fat diet (62% fat, 18.2% protein, and 19.6% carbohydrate; Oriental Yeast Co.) (HFD) for one week before mating, as well as throughout pregnancy and lactation. All the pups were weaned at 5 weeks of age and fed an ND until the age of 8 weeks. Zoledronic acid (ZA, 100 μg/kg) (Sigma Aldrich, St Louis, MO, USA) was intravenously injected just after CaCl_2_ application to induce osteoclast apoptosis and inhibit osteoclast function in the pups [17].

The animals were housed in a room maintained at 22 °C under a 12-h light/dark cycle and provided with drinking water *ad libitum*. At 1, 4, and 8 weeks after CaCl_2_ application, the mice were euthanized by transcardial perfusion under anesthesia induced by isoflurane (2%; 0.2 mL/min). The total number of mice used was 366, not including the preliminary experiments.

### 2.2. Mouse Aneurysm Induction Model

AAAs were induced by the periaortic application of 0.5-M calcium chloride (CaCl_2_), as described in a previous study [18]. Eight-week-old pups were anaesthetized using isoflurane (2%, 0.2 mL/min) by an anesthetic vaporization instrument (PITa-Quark; Sanko Manufacturing Co., Ltd., Saitama, Japan) connected to a nose corn throughout the surgery. The effect of the anesthesia was confirmed via the lack of tail pinch response and closely monitored throughout the procedure with a fine adjustment of isoflurane concentration to maintain the adequate depth of anesthesia. A laparotomy was performed under sterile conditions with the assistance of an operating stereomicroscope. After the abdominal aorta between the left renal artery and the iliac bifurcation was surgically exposed, CaCl_2_-treated cotton gauze was placed directly on the abdominal aorta for 15 min. The gauze was removed, and the intraperitoneal cavity was washed with 0.9% sodium chloride (NaCl) three times before the musculofascial and skin incisions were sutured. In sham-operated mice, 0.9% NaCl was substituted for CaCl_2_. After recovery from anesthesia, monitoring was intensively continued for behavioral signs of postoperative pain with a ready-to-use lidocaine ointment.

### 2.3. Haemodynamic Analysis

Blood pressure and heart rate were measured under conscious and unrestrained conditions using a programmable sphygmomanometer (BP-98A; Softron, Tokyo, Japan) and the tail cuff method [19]. Unanesthetized mice were introduced into a small holder mounted on a thermostatically controlled warming plate and maintained at 37 °C during the measurements. Blood pressure and heart rate were measured three times while the mouse was still in the holder.

### 2.4. Vessel Measurement and Histological Analysis

After transcardial perfusion with 4% paraformaldehyde under physiological pressure following saline perfusion, abdominal aortic tissue was removed by dissecting away the surrounding fatty tissue and scarring adhesions to make the wall of the aorta clearly discernable. After acquiring images, we measured the maximal diameter of cleaned infrarenal aortas using ImageJ software v1.50i (https://imagej.nih.gov/ij/index.html, accessed on 23 August 2021) in all mice from each group. The abdominal aorta (5-mm-long segments), including a portion of the maximal diameter, was excised and embedded in paraffin. Cross-sections (5 μm) of the aortic tissue 2.5~3.0 mm distal to the branch of the left renal artery, including a portion of the maximal diameter, were stained with Elastica van Gieson (EVG) stain and examined via light microscopy.

### 2.5. Immunohistochemical Analysis

Three serial sections (6 µm thick) were prepared from the middle portion of the maximal diameter of AAA and immunohistochemically stained. For the immunological staining of F4/80, anti-F4/80 antibody (1:100, ab6640; Abcam, Cambridge, UK) and Alexa Fluor 488-conjugated secondary antibody (Thermo Fisher Scientific, Waltham, MA, USA) were used. For TNF-α, anti-TNF-α antibody (1:200, ab6671; Abcam) and Alexa Fluor 555-conjugated secondary antibody (Thermo Fisher Scientific) were used. For tartrate-resistant acid phosphate (TRAP), anti-TRAP antibody (1:100, GTX60167; Gene TEX, Irvine, CA, USA) and Alexa Fluor 555-conjugated secondary antibody (Thermo Fisher Scientific) were used. For matrix metalloproteinase (MMP)-9, anti MMP-9 antibody (1:100, ab38898; Abcam) and Alexa Fluor 555-conjugated secondary antibody (Thermo Fisher Scientific) were used. For NFATc1, anti-NFAT2 antibody (1:100, NB300-620; Novus Biologicals, Centennial, CO, USA) and Alexa Fluor 555-conjugated secondary antibody (Thermo Fisher Scientific) were used. The nuclei were labeled using 4′,6-diamidino-2-phenylindole (DAPI) (excitation wavelength: 360nm and fluorescence wavelength 461 nm, 62248; Thermo Fisher Scientific), and the sections were examined using an LSM 510 META confocal microscope (Carl Zeiss, Jena, Germany). Nonimmune immunoglobulin Rabbit IgG, polyclonal Isotype Control was used for negative control of TNF-α, TRAP, and MMP-9. Nonimmune immunoglobulin Rat IgG2b, kappa monoclonal Isotype Control and nonimmune immunoglobulin Mouse IgG1, kappa monoclonal Isotype Control were used for the negative control for F4/80 and NFATc1, respectively. Positive staining was evaluated using ImageJ software v1.50i (https://imagej.nih.gov/ij/index.html, accessed on 23 August 2021). The number of F4/80- or TNF-α-positive stained nuclei was assessed per 3 sections from 5–10 animals from each group. The percentages of TRAP- or MMP-9-positive stained nuclei in F4/80-positive stained nuclei were assessed per 3 sections from 8–10 animals from each group. A total of 4–6 representative images from each animal were randomly chosen and analyzed.

### 2.6. Quantitative Real-Time Polymerase Chain Reaction (qPCR)

Total RNA was extracted from the abdominal aortic tissue using the RNeasy Fibrous Tissue Mini Kit (74704; Qiagen, Hilden, Germany) and was reversely transcribed for the preparation of cDNA with the TAKARA Prime Script RT reagent Kit with gDNA Eraser (RR047A; Takara Bio, Shiga, Japan). Real-time PCR was accomplished using a Thermal Cycler Dice system (Takara Bio) with the KAPA SYBR^®^ FAST Universal qPCR Kit (KK4602; KAPA Biosystems, Wilmington, MA, USA). The dissociation curves were surveyed for the abnormal formation of primer dimers. The threshold cycle (CT) values were normalized to GAPDH, and a comparative expression was calculated using the ΔΔCT method. The data were shown as the gene expression levels relative to those of the controls. The primer pairs are mentioned in Appendix A.

### 2.7. Ex Vivo MMP Activity

We examined the ex vivo MMP activity using a in vivo imaging system (IVIS) according to the previously reported methods [20]. On the day of the experiment, 2-nmol MMPSense 750 FAST (Perkin Elmer, Boston, MA, USA) was administered to all animals via tail vein injection. MMPSense 750 FAST is a targeted fluorescence imaging tracer comprising a MMP recognition sequence, a fluoro-methyl ketone (FMK) leaving group, and a red fluorescent label. FMK-derivatized peptides act as active irreversible inhibitors with no undesirable cytotoxic effects. The probes favorably and permanently bind to active MMPs, causing apoptotic cells to fluoresce. The abdominal aorta was harvested 6 h post-injection, followed immediately by ex vivo aorta imaging using a in vivo imaging system (IVIS) Lumina Series Ⅲ optical imaging platform (PerkinElmer Inc.) employing the red filter sets (excitation, 749 nm; emission, 775 nm longpass). Regions of interest (ROI) encircling the whole organs were manually drawn, and the ensuring signal was calculated in units of scaled counts per second. We cautiously confirmed that the size of the ROIs drawn among the animal samples was constant.

### 2.8. In Vitro Differentiation of BMDMs into Osteoclast-like Macrophages

Bone marrow cells were obtained from femurs and tibias of 8-week-old O-ND and O-HFD mice and cultured in complete medium (α-MEM (Dulbecco’s Modified Eagle’s Medium) supplemented with 10% fetal bovine serum (FBS), 1% penicillin/streptomycin, and 40-ng/mL macrophage-colony stimulating factor-1 (M-CSF)). Nonadherent cells were collected after 24 h and differentiated into bone marrow-derived macrophages (BMDMs) using 40-ng/mL M-CSF for seven days, yielding 98% F4/80^+^ cells. For differentiation into osteoclast-like macrophages, BMDMs were stimulated with 100-ng/mL tumor necrosis factor-α (TNF-α; PEPROTECH, Rocky Hill, NJ, USA) and 40-ng/mL M-CSF for 3 days [17]. For the enhancer of zeste homolog 2 (EZH2) inhibitor treatment or ZA treatment, 10-μM GSK126 (Abcam) or 0.03-μM ZA (Sigma Aldrich) were added to the complete medium during the M-CSF-induced BMDM expansion phase and TNF-α-induced osteoclast differentiation phase. For the classical BMDM polarization assay, BMDMs were stimulated with either 100-U/mL lipopolysaccharide (LPS) (Sigma-Aldrich) or 1-μg/mL elastin-derived peptides (EDP, COSMO BIO, Tokyo, Japan) [21].

### 2.9. Western Blot Analysis

For whole cell protein extraction, BMDMs were harvested and lysed in an extraction buffer (50-mmol/L Tris-HCl (pH 7.5), 150-mmol/L NaCl, 50-mmol/L EDTA, 1% Triton X-100, and protease-phosphatase inhibitor mixture). For nuclear protein extraction, BMDMs were harvested using the NE-PER nuclear and cytoplasmic extraction reagents (78833; Thermo Fisher Scientific) according to the manufacturer’s protocols. The protein samples were subjected to SDS-PAGE and then transferred to membranes that were subsequently incubated with primary antibodies against NFATc1 (1:200, NB300-620; Novus Biologicals, Centennial, CO, USA), NF-κB p65 (1:200, 8242S; Cell Signaling Technology, Danvers, MA, USA), p-NF-κB p65 (1:200, 3033S; Cell Signaling), IRF8 (1:200, A304-028A; Bethyl, Montgomery, TX, USA), β-actin (1:2000, A2228, AC-74; Sigma-Aldrich), LaminB1 (1:2000, 66095-1-IG; COSMO BIO), and α-tubulin (1:2000, T5168, B-5-1-2; Sigma-Aldrich). The immunoreactive proteins were visualized using an ECL-enhanced chemiluminescence detection system (GE Healthcare Life Sciences, Marlborough, MA, USA), followed by exposure to a ChemiDoc XRS Plus imaging system (Bio-Rad Laboratories, Inc., Hercules, CA, USA). The bands were quantified using ImageJ software v1.50i (https://imagej.nih.gov/ij/index.html, accessed on 23 August 2021). Β-actin, LaminB1, or α-tubulin were used as references.

### 2.10. ChIP Assay

A chromatin immunoprecipitation (ChIP) assay was performed using the High-Sensitivity ChIP Kit (ab185913; Abcam) according to the manufacturer’s protocols, with minor modifications. Briefly, BMDMs without TNF-α stimulation were fixed with formaldehyde. After quenching with glycine, the cells were lysed and sonicated. Sonicated chromatin was incubated in a buffer containing antibodies. We used nonimmune immunoglobulin G (Abcam) as the control. After extensive washing, protein–DNA crosslinks were reversed, and the precipitated DNA was treated with proteinase K before phenol chloroform extraction and ethanol precipitation. The anti-trimethyl Histone H3 (Lys27) antibody (Takara Bio) was used for H3K27me3. The primer pairs used are listed in the Appendix A.

### 2.11. Statistical Analysis

Data were expressed as the mean ± standard error of the mean (SEM). The normality of the distribution was assessed by the Shapiro–Wilk test or Anderson–Darling test. Equal variances were assessed by the Fisher test for two groups and the Bartlett test or Brown–Forsythe test for more than three groups. Comparisons were performed using the Mann–Whitney test for two groups or Kruskal–Wallis test for more than three groups if the data were not normally distributed. Student’s *t*-test or analysis of variance (ANOVA), followed by the Tukey–Kramer test to analyze significant differences between the groups. Significant differences among the groups for the dependent variables were detected using two-way ANOVA: maternal diet (ND vs. HFD), ZA treatment, and GSK126 treatment. Otherwise, it was stated in each figure legend. A *p*-value below 0.05 was considered statistically significant. All analyses were performed using GraphPad Prism 8.3.0 for Mac OS (GraphPad Software, LLC, San Diego, CA, USA).

## 3. Results

### 3.1. Maternal HFD Exaggerates AAA Development in Offspring

After 4 and 8 weeks of CaCl_2_ application, the maximum outer diameters of AAA in the male offspring of HFD-fed dams (O-HFD) were significantly larger than those in the male offspring of ND-fed dams (O-ND) (Figure 1A). Consistently, the circumferences of the external elastic membranes were markedly increased in O-HFD compared with those of O-ND at all the time points (Figure 1B). The maternal lipid profile in HFD-fed dams before mating showed significantly higher levels of cholesterol than those in ND-fed dams, almost all of which could be still observed after gestation (Appendix A). The triglyceride level was significantly higher in HFD-fed dams after gestation. The lipid profile in 8-week-old offspring showed that the total cholesterol levels tended to be lower in O-HFD than in O-ND. The low-density lipoprotein cholesterol levels were significantly higher in O-HFD, while the high-density lipoprotein cholesterol levels were lower in O-HFD compared with those in O-ND (Appendix A). There was no difference in the triglyceride levels between the two groups. The body weights of the 8-week-old offspring were compatible between O-HFD and O-ND (Appendix A). Blood pressure and heart rate at 8 weeks after CaCl_2_ application were equivalent between the two groups (Appendix A). In sham-operated mice, there were no differences in the maximum outer diameters and circumferences of the external elastic membranes between the two groups (Appendix A). These findings indicate that augmented AAA development in O-HFD is independent of the hemodynamic variables and is related to the CaCl_2_-induced inflammatory response. We also examined the effect of maternal HFD on AAA development in female offspring. Consistent with the results in the male offspring, the circumferences of external elastic membranes were significantly greater in female O-HFD compared with those in female O-ND at 4 weeks after the CaCl_2_ application (Appendix A). Therefore, subsequent experiments were performed using male offspring only.

### 3.2. Maternal HFD Does Not Affect Accumulation of Classically Activated Macrophage

Interestingly, O-HFD showed a significant expansion of the aneurysm even at an early time point of 1 week after CaCl_2_ application, suggesting that a proinflammatory response was initiated in O-HFD. We therefore examined macrophage accumulation at 1 week after CaCl_2_ application. The mRNA expression levels of F4/80 and CD68 were equivalent between the two groups (Figure 2A). The accumulation of F4/80-positive cells, as well as TNF-α-positive macrophages corresponding to classically activated macrophages, were equivalent between the two groups (Figure 2B). Consistently, the mRNA expression levels of proinflammatory cytokines did not differ between the two groups (Appendix A). These findings suggest that classically activated macrophage accumulation was comparable between the two groups.

### 3.3. TRAP-Positive Macrophage Accumulation Is Enhanced in O-HFD

Osteoclast-like macrophages, one of the subsets of inflammatory macrophages producing high levels of matrix metalloproteinases (MMPs), have been shown to contribute to extracellular matrix degradation and AAA formation [15,17]. Hence, we performed immunostaining for TRAP, a typical marker of osteoclasts. The percentage of TRAP-positive cells in F4/80-positive cells was significantly greater in O-HFD compared to that in O-ND (Figure 3A). Consistently, the percentage of MMP-9-positive cells in F4/80-positive cells was also elevated in O-HFD compared to that in O-ND (Figure 3B). Furthermore, we performed the ex vivo imaging of MMP activity at 1 week after the CaCl_2_ application. The MMP activity in O-HFD was markedly higher than that in O-ND (Figure 3C). We also performed Alizarin red staining in AAA and found that the ratio of positive area to medial area was apparently higher in O-HFD than that in O-ND (Appendix A), suggesting that the aortic calcification in offspring was increased by maternal HFD feeding. These findings suggest that the augmented activity of MMP-9 exerted by osteoclast-like macrophages plays an important role in the development of AAA in the early stages after CaCl_2_ application.

### 3.4. ZA Treatment Eliminates AAA Development in O-HFD

To investigate whether osteoclast-like macrophages were substantially involved in the augmented development of AAA in O-HFD, ZA, which induces osteoclast apoptosis and inhibits osteoclast function, was intravenously injected just after the CaCl_2_ application. The maximum outer diameters in vehicle-treated O-HFD were significantly larger than those in vehicle-treated O-ND at 1 week after the CaCl_2_ application; however, they were comparable between the two groups of ZA-treated mice (Figure 4A). At 4 weeks, ZA treatment inhibited AAA development in O-HFD, with no discernable difference between the two groups (Figure 4B). In a histological analysis, the ZA treatment significantly inhibited the expansion of the circumferences of the external elastic membranes in O-HFD, but they were still larger than those in O-ND at 1 week after the CaCl_2_ application (Figure 4C). However, at week 4, they were equivalent between the two groups of ZA-treated mice (Figure 4D). As it is known that ZA induce apoptosis of the osteoclasts, we examined the effect of ZA on osteoclast-like macrophages in vivo. The number of TRAP-positive cells was scarcely observed in both ZA-treated offspring, resulting in no difference between ZA-treated O-ND and O-HFD (Appendix A). These findings support the theory that osteoclast-like macrophages play a substantial role in the exaggerated AAA development in O-HFD.

### 3.5. TNF-α-Induced Osteoclast-like Macrophage Differentiation Is Enhanced in BMDMs of O-HFD

To examine the effect of maternal HFD on the differentiation of BMDMs into osteoclast-like macrophages, we isolated the bone marrow cells of O-ND and O-HFD and differentiated them into osteoclast-like macrophages using TNF-α following M-CSF priming. The number of TRAP-positive cells was significantly higher in O-HFD after stimulation with TNF-α (Figure 5A,B). Consistently, BMDMs of O-HFD showed higher expression levels of osteoclast-specific genes such as Acp5, Oscar, and Ctsk than those in BMDMs of O-ND (Figure 5C). In line with the in vivo results, the higher expression levels of osteoclast-specific genes in BMDMs of O-HFD were markedly reduced after the ZA treatment (Appendix A). BMDM differentiation into classically activated macrophages did not differ between the two groups (Appendix A).

### 3.6. TNF-α-Induced NFATc1 Expression Is Increased in BMDMs of O-HFD

NFATc1 is a master regulator of osteoclast differentiation [22]. The TNF-α-induced gene and protein expressions of NFATc1 were significantly increased in BMDMs of O-HFD compared with BMDMs of O-ND (Figure 6A,B). To examine whether the nuclear translocation of NFATc1 was augmented in O-HFD, we analyzed the nuclear and cytoplasmic fractions of NFATc1 by immunoblotting. The cytoplasmic protein of NFATc1 was modestly, but not significantly, higher in O-HFD; however, the nuclear protein of NFATc1 did not show any difference between the two groups (Figure 6C). Although immunofluorescence staining for NFATc1 showed that the nuclear translocation of NFATc1 was significantly increased in O-HFD after TNF-α stimulation (Figure 6D,E), the number of nuclear NFATc1-positive cells in O-HFD rarely exceeded a few percent at most, suggesting that the nuclear translocation of NFATc1 has less impact on the enhanced NFATc1 activation in O-HFD.

### 3.7. Expression of Transcriptional Repressor IRF-8 Is Decreased in BMDMs of O-HFD

NF-κB signaling has been reported to be significantly involved in RANKL-induced osteoclast differentiation via the activation of NFATc1. Therefore, we examined NF-κB-p65 activation upon TNF-α stimulation. The phosphorylation of p65 was significantly increased 5 min after TNF-α stimulation and declined gradually. However, no difference could be observed between the two groups (Figure 7A,B). The phosphorylation of p65, as well as p65 protein expression 24 h and 48 h after TNF-α stimulation, was also comparable (Figure 7A,B), suggesting that NF-κB signaling is not likely to be responsible for the enhanced activation of NFATc1 in BMDMs of O-HFD. It is evident that osteoclastogenesis is restrained by transcriptional repressors, such as IRF8, which are basally expressed in osteoclast precursors [23,24,25]. Hence, we examined the mRNA expression levels of transcriptional repressors in the early stage of osteoclast differentiation. The mRNA expression levels of IRF8 were significantly lower in O-HFD than in O-ND before TNF-α stimulation (Figure 7C). While the cytoplasmic fraction of IRF-8 was comparable between the two groups, the nuclear fraction was significantly lower in O-HFD compared with that in O-ND (Figure 7D). Considering the autoamplification of NFATc1 [26], these findings suggest that a lower expression of IRF8 in osteoclast precursors is implicated in the augmented NFATc1 activation after the TNF-α stimulation independent from NF-κB signaling, leading to enhanced differentiation into osteoclast-like macrophages.

### 3.8. EZH2 Inhibitor Restores IRF8 Expression and Ameliorates Differentiation of BMDMs into Osteoclast-like Macrophages in O-HFD

Epigenetic mechanisms underlying the osteoclast differentiation have been previously reported. Fang et al., revealed IRF8 repression by chromatin-based mechanisms in an early stage of osteoclast differentiation [27]. They demonstrated that EZH2 was engaged to the IRF8-promoter region upon RANKL stimulation to deposit the negative histone mark H3K27me3, thereby downregulating the IRF8 expression. Further, the inhibition of EZH2 by the small molecule GSK126 increased the IRF8 expression, accompanied by the downregulation of NFATc1. Therefore, we performed a ChIP assay and found that the histone modification of H3K27me3 at the IRF8 promoter region was significantly higher in the BMDMs of O-HFD before TNF-α stimulation (Figure 8A). To examine the effect of H3K27me3 on IRF8 mRNA expression, BMDMs were treated with EZH2 inhibitor GSK126 before TNF-α stimulation. The treatment with GSK126 significantly augmented the IRF8 mRNA expression in BMDMs of O-HFD, while no discernable difference was observed in the IRF8 mRNA expression in BMDMs of O-ND (Figure 8B). Finally, we examined the effect of GSK126 on the mRNA expression of NFATc1 and TRAP upon TNF-α stimulation. Two days after TNF-α stimulation, the mRNA expressions of NFATc1 and TRAP were significantly increased; however, there was no difference between the two groups (Figure 8C), suggesting that a restored IRF8 expression by GSK126 treatment could reverse the enhanced differentiation of BMDMs into osteoclast-like macrophages in O-HFD.

## 4. Discussion

In this study, we showed for the first time that maternal HFD intake augmented the development of AAA, accompanied by enhanced MMP activity in the periaortic adventitia. The treatment with an osteoclast inhibitor diminished AAA development, suggesting that the accumulation of osteoclast-like macrophages and activity of their related MMPs play a crucial role in enhancing AAA expansion caused by maternal HFD intake. In vitro, the TNF-α-induced differentiation of osteoclast-like macrophages from BMDMs was significantly increased, along with enhanced NFATc1 activity. The basal expression of IRF8, a transcriptional repressor of NFATc1, was significantly lower in BMDMs of O-HFD than in those of O-ND. Furthermore, the histone modification of H3K27me3 was significantly higher at the promoter region of IRF8, and the EZH2 inhibitor treatment restored the IRF8 expression, resulting in no difference in the TNF-α-induced mRNA expressions of NFATc1 and TRAP between the two groups. Our findings provide new insights into the causal effect of maternal HFD on AAA development in offspring, in which the epigenetic alteration of IRF8 expression augments osteoclast-like macrophage differentiation, thereby contributing to enhanced MMP activity.

In the previous studies by Police et al., obesity has been shown to exaggerate AAA development through the augmented inflammation of periaortic adipose tissue [28], and low-fat diet-induced weight loss inhibited the progression of established AAA [29]. We have previously shown that maternal HFD exaggerated the insulin resistance of HFD-fed offspring through an augmented inflammatory response in epidydimal adipose tissue, which was not observed in the offspring before HFD feeding [11]. Considering that the body weights of 8-week-old offspring were compatible between O-HFD and O-ND (Appendix A), obesity and the subsequent periaortic adipose tissue inflammation are not likely to be responsible for augmented AAA development in O-HFD. Hypercholesterolemia has also been shown to promote AAA development through the macrophage-mediated inflammatory response, including MMP activation [30]. We examined the lipid profile in the offspring and found that the total cholesterol levels were modestly lower in O-HFD than those in O-ND (Appendix A). In contrast, the low-density lipoprotein cholesterol levels were significantly higher in O-HFD than those in O-ND, while the high-density lipoprotein cholesterol levels were lower in O-HFD than those in O-ND. Considering that the blood cholesterol levels were much lower than those in high-cholesterol diet-fed apoE-deficient mice and wild-type mice carrying gain-of-function mutations of PCSK9 [31], hypercholesterolemia is not likely to contribute to the augmented AAA development in O-HFD.

The underlying mechanisms of AAA development have been intensively investigated over the past decades; however, a definite medical therapy to prevent AAA progression has not yet been established [13,14]. Based on the fact that the inflammatory response and consequent ECM degradation by MMPs play a crucial role in the initiation and progression of AAA development [32,33], a distinct phenotype of macrophages, i.e., osteoclasts exhibiting high MMP activity, has recently emerged as osteoclast-like macrophages [15,17]. Takei et al., demonstrated that osteoclast-like macrophages expressing TRAP, a typical marker of osteoclasts, could be observed in the perivascular lesions of human AAA tissue [17]. They also showed that osteoclast-like macrophages play a substantial role in elastase-induced AAA, as well as in angiotensin II-infused dissecting aneurysm in a murine model [34], suggesting that osteoclast-like macrophages could be a novel therapeutic target for preventing AAA development. We also observed that the maternal HFD intake augmented AAA development in the offspring, which was diminished by the treatment with ZA; however, the ZA treatment did not affect the AAA development in O-ND. Besides the lowering effect on the calcium levels, ZA has been shown to inhibit the mevalonate pathway, which is crucially involved in the synthesis of GTP-binding proteins such as Ras, Rho, Rac, and Rab, thereby leading to the induction of apoptosis in osteoclasts [35,36]. We performed Alizarin red staining to detect the calcium deposit in the CaCl_2_-induced AAA model and found that positive staining was much lower compared with that in the previous study, in which CaCl_2_ plus phosphate-buffered salts (PBS) was applied to develop the AAA model [17]. These findings suggest that osteoclast-like macrophages were less prevalent in the CaCl_2_-induced AAA model, resulting in no effect of the ZA treatment on the AAA development in O-ND. These findings not only support the notion that osteoclast-like macrophages could be a therapeutic target for AAA development but also raises the possibility that the epigenetic reprogramming of differentiation of osteoclast-like macrophages could be a potential therapeutic strategy for arresting AAA development.

Yokota et al., have recently shown that the TNF-α treatment of human peripheral monocytes induced abundant TRAP-positive cells despite only a few multinucleated cells compared with the RANKL treatment [37]. In contrast, TNF-α-induced osteoclast-like macrophages highly expressed MMP3 mRNA compared with RANKL-induced osteoclasts. Therefore, osteoclast-like macrophages are not synonymous with osteoclasts, and it is important to characterize their phenotype and relative significance in the pathogenesis of chronic inflammatory diseases. The differentiation of osteoclast-like macrophages has been shown to be induced by inflammatory cytokines such as TNF-α, which are produced and released from various kinds of cells involved in the pathogenesis of AAA development [16,38,39,40]. Therefore, we cannot exclude the possibility that inflammatory cytokines released from these cells may affect the differentiation of osteoclast-like macrophages. The precise mechanisms of osteoclast-like macrophage differentiation in vivo need to be investigated in future studies.

NFATc1 has an essential role in osteoclast differentiation and is also a transcriptional master regulator in osteoclast-like macrophage differentiation [17]. Upon stimulation by inflammatory cytokines like TNF-α, the NF-κB signaling pathway via Toll-like receptors activates NFATc1, leading to the expression of osteoclast-specific genes. However, the TNF-α-induced phosphorylation of NF-κB p65 was comparable between the two groups. Besides stimulation by this pathway, several negative regulators have been reported to be involved in NFATc1 activation through their downregulation after the inflammatory response [41,42]. Notably, Zhao et al., reported that IRF8-deficient osteoclast precursors are more likely to differentiate into osteoclasts upon stimulation by TNF-α, as well as RANKL [24]. Further, IRF8-deficient mice showed osteoporosis accompanied by an increased number of osteoclasts. The study showed that IRF8 suppresses the expression and function of NFATc1 by physically interacting with NFATc1 and binding to its target genes, as well as to its promoter site [26]. We observed that the gene and protein expression levels of IRF8 in the BMDMs of O-HFD were significantly lower than those in BMDMs of O-ND and that the TNF-α-induced NFATc1 expression was significantly augmented in O-HFD. These findings suggest that a lower basal expression of IRF8 substantially contributes to an increased TNF-α-induced NFATc1 activity and subsequent differentiation of BMDMs into osteoclast-like macrophages in O-HFD.

The epigenetic regulation of osteoclast differentiation has been reported via both positive and negative regulators [26,43,44,45,46,47,48]. The epigenetic modulation of IRF8 expression has also been reported to affect physiological and pathological bone homeostasis. Nishikawa et al., first reported that IRF8 expression was downregulated by DNA methyltransferase3a (Dnmt3a), DNA methyltransferase activated by RANKL in the middle stage of osteoclast differentiation [49]. Dnmt3a-deficient osteoclast precursor cells do not differentiate efficiently into osteoclasts. However, they also showed that the overexpression of Dnmt3a in non-RANKL-treated BMDMs did not affect both DNA methylation at the IRF8 gene locus and the IRF8 mRNA expression level, suggesting that the expression of Dnmt3a alone was not sufficient to suppress IRF8 expression in osteoclast precursors. Although IRF8 is a direct target of Dnmt3a-mediated DNA methylation in the middle stage of osteoclast differentiation, the augmented expression of Dnmt3a is not likely to substantially contribute to a reduced IRF8 expression in the BMDMs of O-HFD. Fang et al., also demonstrated that IRF8 expression was negatively regulated by the epigenetic repressor H3K27me3 in osteoclast precursors and that the EZH2 inhibitor treatment increased the IRF8 expression accompanied by a reduction in NFATc1 activity and expression of osteoclast-specific genes [27]. We performed a ChIP assay and found that BMDMs of O-HFD had a higher enrichment of H3K27me3 at the IRF8 promoter region and that the treatment with the EZH2 inhibitor restored the IRF8 mRNA expression in O-HFD to the same level as in O-ND, resulting in no difference in TNF-α-induced NFATc1 and TRAP expressions between the two groups. These findings suggest that the maternal HFD intake promotes the differentiation of osteoclast-like macrophages by reducing the basal expression of IRF8 through the histone modification of H3K27me3.

Recently, Adamic et al., demonstrated the precise mechanisms of EZH2-mediated modification in RANKL-induced osteoclast differentiation [48]. Consistent with previous reports [27], they showed that the EZH2 inhibitor GSK126 effectively blocked the EZH2 methyltransferase activity without altering the EZH2 mRNA and protein levels and prevented the downregulation of osteoclast differentiation inhibitory factors such as MafB, IRF8, and Arg1 [48]. They further demonstrated that the EZH2 mRNA and protein levels were quickly upregulated after RANKL stimulation and that the repressor H3K27me3 marks were significantly increased at the promoter site of the negative regulators for osteoclastogenesis. In contrast, the treatment with GSK126 just before RANKL stimulation did not affect the subsequent osteoclast differentiation. Considering that BMDMs of O-HFD showed a higher enrichment of H3K27me3 at the IRF8 promoter site before TNF-α stimulation and that the treatment with GSK126 significantly restored the IRF8 mRNA expression, it seems that the EZH2 activity in O-HFD is higher than that in O-ND. Consistent with our hypothesis, Adamic et al., showed that the overexpression of EZH2 using lentivirus transduction in the primary osteoclast precursors enhanced the RANKL-driven decrease in MafB expression accompanied by elevated cellular H3K27me3 marks, leading to a significantly higher formation of multinucleated osteoclasts. Although they did not report IRF8 expression levels in EZH2-overexpressing osteoclast precursor cells, it is plausible that augmented EZH2 activity in O-HFD reduced the expression of the negative regulator IRF8, thereby enhancing the TNF-α-induced NAFTc1 expression and subsequent osteoclast-like macrophage differentiation.

Numerous studies have been conducted on epigenetic modulations of the genes implicated in AAA development, most of which involve inflammatory cytokines derived from classically activated macrophages and mitogenic signaling in vascular smooth muscle cells (VSMCs) [50,51,52,53,54,55]. Epigenetic modulations by DNA methylation, histone acetylation, and ncRNA have been extensively investigated; however, the efficacy of their clinical applications, including histone deacetylase (HDAC) inhibitors, has not been fully established. Recently, Lino Cardenas et al., reported that EZH2 inhibitor treatment reduced the expansion of thoracic aortic aneurysms in Fbn1(C1039G/+) mice, referred to as Marfan mice, by improving the cytoskeletal architecture, along with restoring the SM22α expression [56]. EZH2 inhibitors have emerged as a promising therapeutic strategy for cancer patients [57,58]. EZH2 activity has been shown to be upregulated in a variety of malignancies, such as breast, prostate, and bladder cancers [57,58], preferentially complicated by osteolytic bone metastasis, in which augmented osteoclast differentiation plays a crucial role [59]. Further, cancer patients are well-known to have a substantially higher risk of CVDs [60,61]. Interestingly, the standardized mortality ratio for aortic rupture/aneurysms has been shown to be the highest amongst the six types of cardiovascular diseases, including heart disease, during follow-ups [61]. It is therefore clinically relevant to assess the protective effects of EZH2 inhibitors on AAA expansion, as well as tumor progression, in cancer patients. In this sense, our findings suggest a novel approach regarding a practical therapeutic strategy for preventing AAA progression, especially in cancer patients.

Although the maternal intake of HFD during pregnancy increases the risk of various kinds of chronic inflammatory diseases in the offspring, there is hardly any research focusing on AAA development. On the other hand, the maternal HFD-associated risk for cancer has been reported. de Assis et al., showed that the maternal HFD intake during pregnancy increases the mammary tumor incidence in daughters and granddaughters. However, the mammary mRNA levels of de novo methyltransferases Dnmt3a and Dnmt3b were not altered, suggesting the involvement of other epigenetic mechanisms in mediating the multigenerational effects of maternal HFD [62]. Benesh et al., also showed that the maternal HFD intake induces hyperproliferation in the prostate, accompanied by attenuated activity of the phosphatase and tensin homolog, a cardinal prostate cancer tumor suppressor, suggesting that the maternal HFD intake is a risk factor for prostate cancer in adult offspring [63]. The effect of maternal HFD intake on bone homeostasis has also been extensively investigated; however, most of the studies have focused on osteoblastic cell differentiation and proliferation during normal skeletal development in the offspring [64,65]. In pathological conditions such as osteoporosis and osteolytic bone metastasis, osteoclastogenesis. as well as osteogenesis, play crucial roles [59]. Cancer patients are well-known to be coincident with osteoporosis, although it is generally thought to be due to the deleterious effects of cancer-specific therapies [66]. Cumulatively, our findings suggest that the maternal HFD-induced augmentation of osteoclastogenesis through enhanced EZH2 activity may be a common epigenetic basis for AAA progression and osteolytic bone destruction in specific types of cancers and may facilitate a further understanding of osteoclast-associated chronic inflammatory disorders triggered by the maternal HFD intake.

## 5. Conclusions

Our study demonstrates that the maternal HFD intake exaggerates MMP activation in the periaortic adventitia of offspring and subsequently accelerates AAA development, which is abolished by the treatment with the pharmacological inhibition of osteoclastogenesis. Furthermore, the in vitro differentiation of osteoclast-like macrophages from BMDMs was significantly augmented by the maternal HFD intake, accompanied by a decreased IRF8 expression and subsequently enhanced activation of NFATc1. Furthermore, the enrichment of H3K27me3 at the IRF8 promoter region plays a crucial role in the attenuated IRF8 expression. These findings support the hypothesis that augmented differentiation and the accumulation of osteoclast-like macrophages play a critical role in maternal HFD-mediated AAA development in offspring and offer new insights into the underlying mechanism of maternal HFD-related CVD development by focusing on the modulation of macrophage function.

## Figures and Tables

**Figure 1 cells-10-02224-f001:**
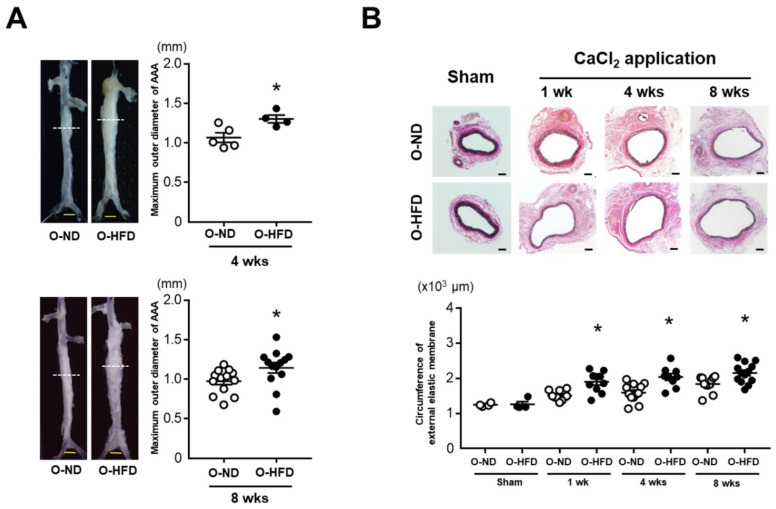
**Maternal HFD exaggerates the development of CaCl_2_-induced AAA in male offspring**. (**A**) Representative photographs and quantitative measurements of the maximum outer diameters at 4 and 8 weeks after the CaCl_2_ application. A dotted line indicates the level of the maximum outer diameter. Values represent the mean ± SEM for 5 O-ND and 4 O-HFD mice at 4 weeks and 13 O-ND and 13 O-HFD mice at 8 weeks. * *p* < 0.05 vs. O-ND at the corresponding sampling point; Student’s *t*-test. O-ND, offspring of ND-fed dam; O-HFD, offspring of HFD-fed dam. Scale bar = 1 mm. (**B**) Representative photographs and quantitative measurements of the circumferences of the external elastic membrane before and at 1, 4, and 8 weeks after the CaCl_2_ application. Values represent the mean ± SEM for 4 O-ND and 4 O-HFD mice before the application, 10 O-ND and 10 O-HFD mice at 1 week, 12 O-ND and 9 O-HFD mice at 4 weeks, and 11 O-ND and 13 O-HFD mice at 8 weeks after the CaCl_2_ application. * *p* < 0.05 vs. O-ND at the corresponding sampling point after the CaCl_2_ application; one-way ANOVA with the Tukey–Kramer post hoc test. O-ND, offspring of ND-fed dam; O-HFD, offspring of HFD-fed dam. Scale bar = 100 μm.

**Figure 2 cells-10-02224-f002:**
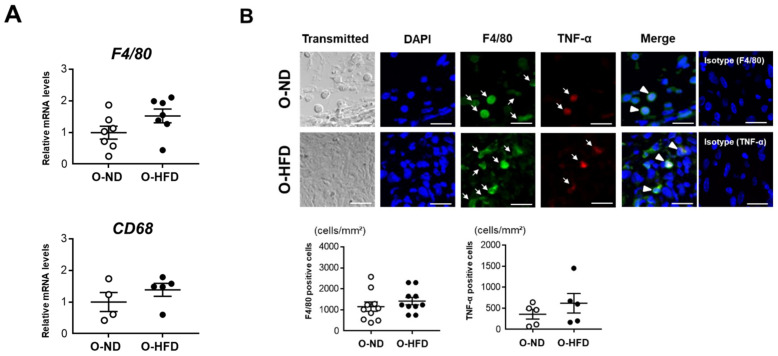
**Maternal HFD does not affect the accumulation of macrophages and their TNF-α expression**. (**A**) Quantitative PCR analysis of mRNA expression levels of F4/80 and CD68 in AAA at 1 week after CaCl_2_ application. Values represent the mean ± SEM relative to O-ND. Each group consisted of 7 O-ND and 7 O-HFD samples for F4/80 and 4 O-ND and 5 O-HFD samples for CD68. Statistical analysis was made by the Student’s *t*-test. O-ND, offspring of ND-fed dam; O-HFD, offspring of HFD-fed dam. (**B**) Representative fluorescent images of F4/80-positive cells and TNF-α-positive macrophages and a quantitative analysis in AAA from O-ND and O-HFD mice at 1 week after CaCl_2_ application. Values are the mean ± SEM for 10 O-ND and 10 O-HFD mice for F4/80-positive cells and 5 O-ND and 5 O-HFD mice for TNF-α-positive macrophages. Arrows indicate positively stained cells for TNF-α or F4/80. Arrowheads indicate TNF-α/F4/80 double-positive cells. Statistical analysis was made by the Student’s *t*-test. O-ND, offspring of ND-fed dam; O-HFD, offspring of HFD-fed dam; TNF-α, tumor necrosis factor-α. Scale bar = 25 μm.

**Figure 3 cells-10-02224-f003:**
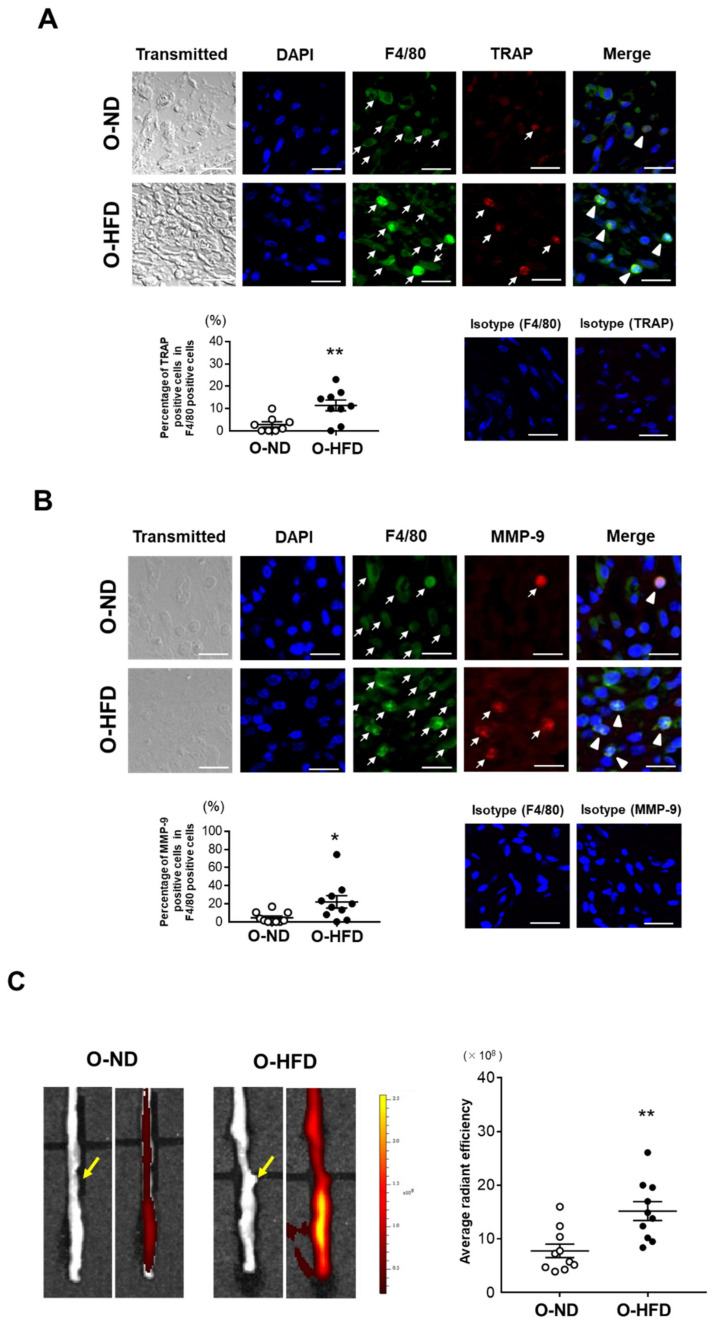
**Maternal HFD augments the accumulation of MMP-9-positive macrophages**. (**A**) Representative fluorescent images of TRAP and F4/80-positive cells and a quantitative analysis of the percentage of TRAP-positive cells in the total number of F4/80-positive cells in AAA from O-ND and O-HFD mice at 1 week after CaCl_2_ application. Arrows indicate positively stained cells for TRAP or F4/80. Arrowheads indicate TRAP/F4/80 double-positive cells. Values are the mean ± SEM for 8 O-ND and 9 O-HFD mice. ** *p* < 0.01 vs. O-ND; Student’s *t*-test. O-ND, offspring of ND-fed dam; O-HFD, offspring of HFD-fed dam. TRAP, tartrate-resistant acid phosphatase. Scale bar = 25 μm. (**B**) Representative fluorescent images of MMP-9 and F4/80-positive cells and quantitative analysis of the percentage of MMP-9-positive cells in the total number of F4/80-positive cells in AAA from O-ND and O-HFD mice at 1 week after CaCl_2_ application. Arrows indicate positively stained cells for MMP-9 or F4/80. Arrowheads indicate MMP-9/F4/80 double-positive cells. Scale bar = 25 μm. Values are the mean ± SEM for 10 O-ND and 10 O-HFD mice. * *p* < 0.05 vs. O-ND; Mann–Whitney test. O-ND, offspring of ND-fed dam; O-HFD, offspring of HFD-fed dam; MMP-9, matrix metalloproteinase-9. (**C**) Representative ex vivo images of AAA and quantitative measurement of the radiant efficiency corresponding to MMP activity. Arrows indicate the origin of the left renal artery. Values are the mean ± SEM for 10 O-ND and 10 O-HFD mice. ** *p* < 0.01 vs. O-ND; Student’s *t*-test. O-ND, offspring of ND-fed dam; O-HFD, offspring of HFD-fed dam; MMP, matrix metalloproteinase.

**Figure 4 cells-10-02224-f004:**
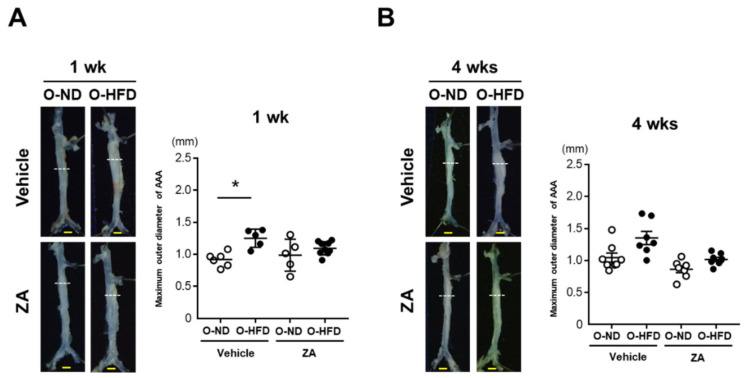
**ZA treatment diminishes the exaggerated AAA development in O-HFD.** (**A**,**B**) Representative photographs and quantitative measurements of the maximum outer diameters at 1 and 4 weeks after CaCl_2_ application with and without ZA treatment. A dotted line indicates the level of maximum outer diameter. Values represent the mean ± SEM for 6 O-ND, 5 O-HFD, 5 ZA-treated O-ND, and 9 ZA-treated O-HFD mice at 1 weeks, as well as 8 O-ND, 7 O-HFD, 7 ZA-treated O-ND, and 8 ZA-treated O-HFD mice at 4 weeks. * *p* < 0.05 vs. O-ND; two-way ANOVA with the Tukey–Kramer post hoc test (**A**) and Kruskal–Wallis test (**B**). O-ND, offspring of ND-fed dam; O-HFD, offspring of HFD-fed dam. ZA, zoledronic acid. Scale bar = 1 mm. (**C**,**D**) Representative photographs and quantitative measurements of the circumferences of the external elastic membrane at 1 and 4 weeks after CaCl_2_ application with and without ZA treatment. Values represent the mean ± SEM for 6 O-ND, 5 O-HFD, 5 ZA-treated O-ND, and 9 ZA-treated O-HFD mice at 1 week, as well as 8 O-ND, 7 O-HFD, 7 ZA-treated O-ND, and 8 ZA-treated O-HFD mice at 4 weeks.** *p* < 0.01 vs. O-ND at the corresponding sampling point.^##^
*p* < 0.01 vs. O-HFD. ^¶¶^
*p* < 0.01 vs. ZA-treated O-ND; two-way ANOVA with the Tukey–Kramer post hoc test (C, D). O-ND, offspring of ND-fed dam; O-HFD, offspring of HFD-fed dam. ZA, zoledronic acid. Scale bar = 100 μm.

**Figure 5 cells-10-02224-f005:**
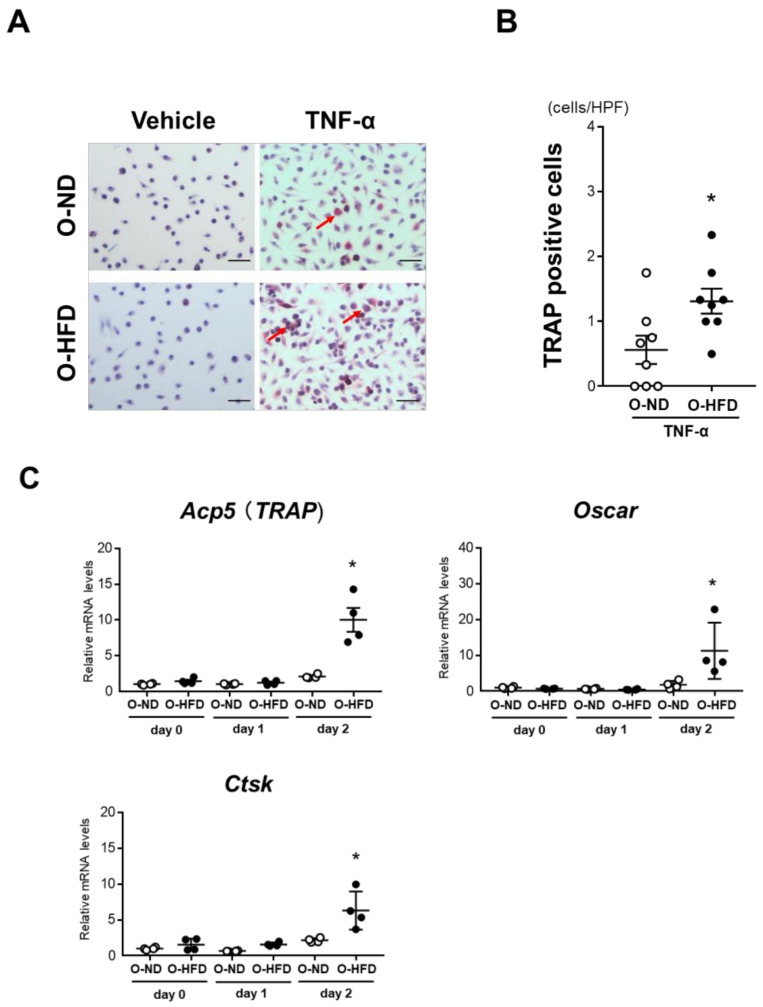
**TNF-α-induced osteoclast-like macrophage differentiation is augmented in BMDMs of O-HFD.** (**A**,**B**) Representative photographs and quantitative measurements of TRAP-positive cells with and without TNF-α stimulation. Values represent the mean ± SEM for 8 O-ND, 8 O-HFD, 8 TNF-α-stimulated O-ND, and 8 TNF-α-stimulated O-HFD BMDMs. * *p* < 0.05 vs. O-ND; Student’s *t*-test. O-ND, offspring of ND-fed dam; O-HFD, offspring of HFD-fed dam. TRAP, tartrate-resistant acid phosphatase; TNF-α, tumor necrosis factor-α. Scale bar = 50 μm. (**C**) Quantitative PCR analysis of the mRNA expression levels of osteoclast differentiation-related genes. Values represent the mean ± SEM relative to O-ND. Each group consisted of 4 O-ND and 4 O-HFD BMDMs at the corresponding sampling point. * *p* < 0.05 vs. O-ND at day 2 after TNF-α stimulation; Mann–Whitney test (Acp5 and Oscar) and Welch’s *t*-test (Ctsk). TNF-α, tumor necrosis factor-α; Acp5, tartrate-resistant acid phosphatase type 5; Ctsk, cathepsin K. O-ND, offspring of ND-fed dam; O-HFD, offspring of HFD-fed dam.

**Figure 6 cells-10-02224-f006:**
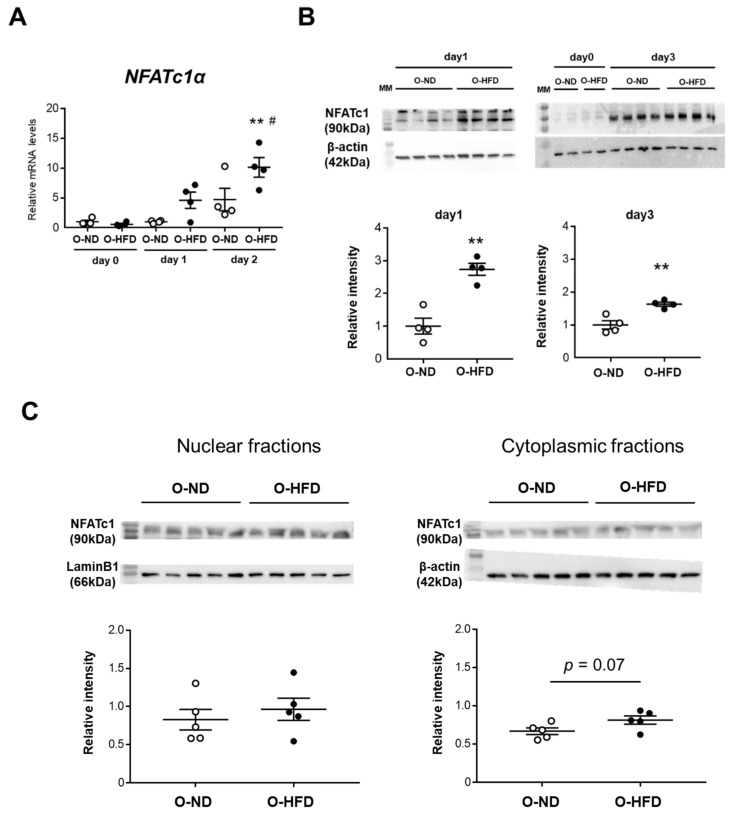
**TNF-α-induced gene and protein expressions of NFATc1 are augmented in BMDMs of O-HFD.** (**A**) Quantitative PCR analysis of the NFATc1 mRNA expression levels. Values represent the mean ± SEM relative to O-ND. Each group consisted of 4 O-ND and 4 O-HFD BMDMs at the corresponding sampling point. ** *p* < 0.01 vs. O-HFD before TNF-α stimulation. ^#^ *p* < 0.01 vs. O-ND (day 2); Kruskal–Wallis test. O-ND, offspring of ND-fed dam; O-HFD, offspring of HFD-fed dam. NFATc1, nuclear factor of activated T cells, cytoplasmic 1. (**B**) Protein expression levels of NFATc1 after TNF-α stimulation. Each group consisted of 4 O-ND and 4 O-HFD BMDMs. ** *p* < 0.01 vs. O-HFD at the corresponding sampling point; Student’s *t*-test. O-ND, offspring of ND-fed dam; O-HFD, offspring of HFD-fed dam. NFATc1, nuclear factor of activated T cells, cytoplasmic 1; TNF-α, tumor necrosis factor-α. (**C**) Representative Western blot of cytoplasmic and nuclear NFATc1 in BMDMs at 1 day after TNF-α stimulation, and a quantitative analysis of the protein expression. Values are the mean ± SEM for 5 O-ND and 5 O-HFD BMDMs. Student’s *t*-test. O-ND, offspring of ND-fed dam; O-HFD, offspring of HFD-fed dam. (**D**) Representative fluorescent images of nuclear translocated NFATc1 after TNF-α stimulation. Scale bar = 10 μm. (**E**) Representative photographs and quantitative measurements of relative nuclear NFATc1-positive cells with TNF-α stimulation. Values represent the mean ± SEM for 5 TNF-α-stimulated O-ND and 5 TNF-α-stimulated O-HFD BMDMs. * *p* < 0.05 vs. O-ND upon TNF-α stimulation; Welch’s *t*-test. O-ND, offspring of ND-fed dam; O-HFD, offspring of HFD-fed dam. NFATc1, nuclear factor of activated T cells, cytoplasmic 1; TNF-α, tumor necrosis factor-α. Scale bar = 20 μm.

**Figure 7 cells-10-02224-f007:**
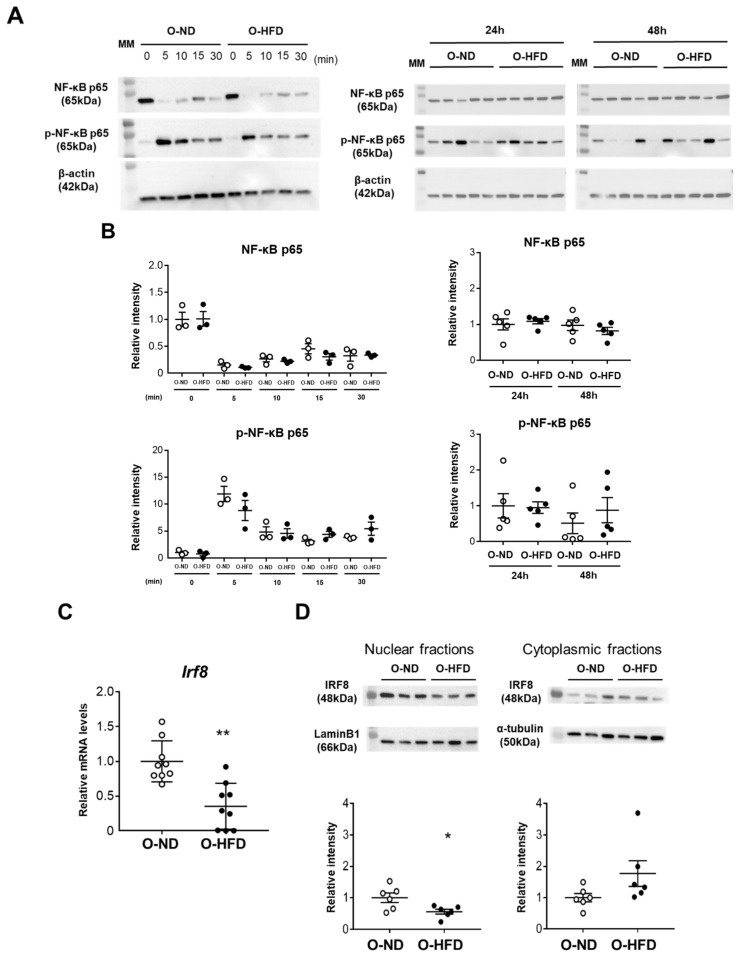
**Expression of IRF8 is attenuated in BMDMs of O-HFD via the histone modification of H3K27me3.** (**A,B**) Representative Western blot of NF-κB-p65 in BMDMs upon TNF-α stimulation, and a quantitative analysis of the protein expression. Values are the mean ± SEM for 3 O-ND and 3 O-HFD BMDMs at each time point until 30 min and for 5 O-ND and 5 O-HFD BMDMs at 24 h and 48 h; two-way ANOVA with the Tukey–Kramer post hoc test. O-ND, offspring of ND-fed dam; O-HFD, offspring of HFD-fed dam. NF-κB, nuclear factor-κB; TNF-α, tumor necrosis factor-α. (**C**) Quantitative PCR analysis of the IRF8 mRNA expression levels. Values represent the mean ± SEM relative to O-ND. Each group consisted of 9 O-ND and 9 O-HFD BMDMs. ** *p* < 0.01 vs. O-ND; Student’s *t*-test. O-ND, offspring of ND-fed dam; O-HFD, offspring of HFD-fed dam. IRF8, interferon regulatory factor 8. (**D**) Representative Western blot of nuclear and cytoplasmic IRF8 in BMDMs, and a quantitative analysis of the protein expression. Values are the mean ± SEM for 6 O-ND and 6 O-HFD BMDMs. * *p* < 0.05 vs. O-ND; Student’s *t*-test (nuclear IRF8) and Welch’s *t*-test (cytoplasmic IRF8). O-ND, offspring of ND-fed dam; O-HFD, offspring of HFD-fed dam. IRF8, interferon regulatory factor 8.

**Figure 8 cells-10-02224-f008:**
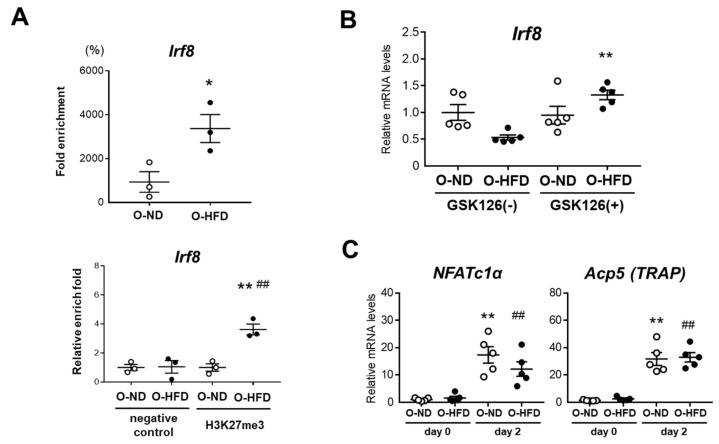
H3K27me3 marks are enhanced at IRF8 promoter region and their inhibition attenuates differentiation into osteoclast-like macrophages in O-HFD (**A**) ChIP assays for H3K27me3 at the IRF8 promoter in BMDMs. Values represent the mean ± SEM for 3 O-ND and 3 O-HFD BMDMs. * *p* < 0.05 vs. O-ND; Student’s *t*-test (fold enrichment). ** *p* < 0.01 vs. O-ND (negative control), ^##^
*p* < 0.01 vs. O-ND (H3K27me3); two-way ANOVA with the Tukey–Kramer post hoc test (qPCR). O-ND, offspring of ND-fed dam; O-HFD, offspring of HFD-fed dam. IRF8, interferon regulatory factor 8. (**B**) Quantitative PCR analysis of the IRF8 mRNA expression levels with and without EZH2 inhibitor GSK126. Values represent the mean ± SEM for 5 O-ND and 5 O-HFD BMDMs. ** *p* < 0.01 vs. O-ND; Kruskal–Wallis test. O-ND, offspring of ND-fed dam; O-HFD, offspring of HFD-fed dam. IRF8, interferon regulatory factor 8; EZH2, enhancer of zeste homolog 2. (**C**) Quantitative PCR analysis of the mRNA expression levels of NFATc1 and TRAP after TNF-α stimulation in GSK126-treated BMDMs. Values represent the mean ± SEM for 5 O-ND and 5 O-HFD BMDMs. ** *p* < 0.01 vs. O-ND. ^##^ *p* < 0.01 vs. O-HFD; one-way ANOVA with the Tukey–Kramer post hoc test. O-ND, offspring of ND-fed dam; O-HFD, offspring of HFD-fed dam. NFATc1, nuclear factor of activated T cells, cytoplasmic 1; TRAP, tartrate-resistant acid phosphatase; TNF-α, tumor necrosis factor-α.

## Data Availability

Not applicable.

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
