# Peer review of "Maternal High-Fat Diet Promotes Abdominal Aortic Aneurysm Expansion in Adult Offspring by Epigenetic Regulation of IRF8-Mediated Osteoclast-like Macrophage Differentiation"

_cells, 2021, doi:10.3390/cells10092224_

Round 1
Reviewer 1 Report
Saburi and co-authors present their work on the role of a high fat diet in abdominal aortic aneurysm expansion by means of epigenetic regulation of IRF8-mediated differentiation. Here are some of my comments:
Line 80 - Zoledernoic acid is typically used for lowering calcium levels. Since the AAA model is created using calcium chloride, this step seems contradictory. Please explain how this treatment does not affect the normal course of aneurysm development by CaCl2?
Line 86 - did the authors actually use 366 animals in this study?
Line 119 - What does the middle portion entail? Please describe clearly what criterion was utilized by the authors to constitute an aneurysmal event. What were the parametric controls that we set to classify an animal as aneurysmal?
Figure 1 - How was the exact point of measurement of "maximum outer diameter"? It is unclear how do the O-ND specimens shown here in Fig. 1 (A) qualify as an aneurysm?
MMP-9 IHC staining is typically not conclusive. Did the authors perform gelatin zymography to confirm their MMP observations? Without zymography the MMP data shown here is not concrete enough to draw conclusions.
Author Response
Please find the attached file below.

Reviewer 2 Report
The study by Saburi et al. investigated the impact of maternal high fat diet (HFD) on abdominal aortic aneurysm (AAA) development. After mice were fed an HFD or normal diet one week prior to mating, AAA was induced in eight-week-old pups using calcium chloride application. Male offspring of HFD-fed dams showed an enlargement in AAA. Moreover, TRAP and MMP activity were increased and inhibition of osteoclastogenesis abolished the exaggerated AAA. The authors concluded that maternal HFD augmented aortic expansion in adult offspring. The exaggerated osteoclast-like macrophages accumulation and increased MMP activity suggested the possibility of macrophages skewing towards osteoclast-like cells via epigenetic reprogramming.
Comments:
- The authors studied the development of AAA histologically for 8 weeks but they did not follow the dynamic of aortic diameters in vivo. Why?
- At which localization in aorta did you compare the diameters?
- Sham operated animals are missing.
- How is it possible to measure the aortic diameter histologically before application of Calcium chloride (Figure 1)?
- The authors suggest that the inflammatory mediators play a role by HFD-induced vascular remodeling in the offsprints. Could you provide the data about the regulation of the inflammatory parameters in serum?
- The information about pharmacological inhibition of osteoclastogenesis is missing.
- Too much data that is difficult to interpret. It would be important to present the main findings and describe them in a schema.
- The discussion is too long and does not focus on the main findings. Treatment with ZM is even not mentioned in the conclusion.
- Previous studies that investigated the aortic dilatation in mice by HFD and treatments should be mentioned.
- Line 116:…mm should be µm.
Author Response
Please find the attached file below.

Reviewer 3 Report
Saburi et al. reported that maternal HFD-induced reprogramming of macrophages in the offspring contributes to the enhanced development of abdominal aortic aneurysm and that therapeutic targeting of the epigenetic modifications in the macrophage phenotype could potentially remediate and prevent abdominal aortic aneurysm development. Overall, the study is well designed and the outcomes are appropriate. The experimental section is descriptive with the detailed discussion of the results.
Only a few minor typos and editing errors:
- Some acronyms should have full name at the first appearance, such as TNF, NFATc1, EZH2, and BM in Abstract and Dnmt (line 566). At line 603, HDAC only written once? So, no acronym is needed.
- To cite one article, you may write “last name” et al,. There is no need to add first name. To be consistent of the writing, please correct some in text citations, such as in line 568, 604, 621, and 625.
Author Response
Please find the attached file below.

Round 2
Reviewer 2 Report
The authors improved the manuskript and answered the questions. The paper can be accepted for publication.